# Student’s Inventory of Professionalism (SIP): A Tool to Assess Attitudes towards Professional Development Based on Palliative Care Undergraduate Education

**DOI:** 10.3390/ijerph16244925

**Published:** 2019-12-05

**Authors:** Antonio Noguera, María Arantzamendi, Jesús López-Fidalgo, Alfredo Gea, Alberto Acitores, Leire Arbea, Carlos Centeno

**Affiliations:** 1Symptom Control and Palliative Medicine Department, Clínica Universidad de Navarra, 31008 Pamplona (Navarra), Spain; ccenteno@unav.es; 2ATLANTES Research Programme, Institute for Culture and Society, University of Navarra, 31008 Pamplona (Navarra), Spain; marantz@unav.es (M.A.); fidalgo@unav.es (J.L.-F.); acancela@alumni.unav.es (A.A.); 3Instituto de investigación sanitaria de Navarra (IdiSNA), 31009 Pamplona (Navarra), Spain; 4Epidemiology Department, Faculty of Medicine, University of Navarra, 31008 Pamplona (Navarra), Spain; ageas@unav.es; 5Radiation Oncology Department, Clínica Universidad de Navarra, 31008 Pamplona (Navarra), Spain; larbea@unav.es; 6Medical Education Unit. Faculty of Medicine, University of Navarra, 31008 Pamplona (Navarra), Spain

**Keywords:** clinical education, professional development, assessment, medical students, undergraduate education, palliative care education

## Abstract

*Introduction*: Quality medical education, centered on a patient’s needs, is crucial to develop the health professionals that our society requires. Research suggests a strong contribution of palliative care education to professionalism. The aim of this study was to design and validate a self-report inventory to measure student’s professional development. *Method*: Sequential exploratory strategy mixed method. The inventory is built based on the themes that emerged from the analysis of four qualitative studies about nursing and medical students’ perceptions related to palliative care teaching interventions (see Ballesteros et al. 2014, Centeno et al. 2014 and 2017, Rojí et al. 2017). The structure and psychometrics of the inventory obtained is tested in two different surveys with two different groups of medical students. Inventory reliability and construct validity are tested in the first survey group. To verify the inventory structure, a confirmatory factor analysis is performed in a second survey group. *Results*: The inventory has 33 items and seven dimensions: a holistic approach, caring for and understanding the patient, personal growth, teamwork, decision-making, patient evaluation, and being a health care professional. Cronbach’s-alpha was 0.73–0.84 in all seven domains, ICC: 0.95. The confirmatory factor analysis comparative fit index (CFI) was 1 with a standardized root mean square Index 0.088 (SRMR) and obtained a 0.99 goodness-of-fit R-square coefficient. Conclusions: this new inventory is grounded on student’s palliative care teaching experiences and seems to be valid to assess student’s professional development.

## 1. Introduction

Health faculties should provide future professionals with the necessary competencies that society, patients, and the very profession acknowledge as their professional role. Medical education is moving again towards patient-centered models of care that assess the holistic experience of patient’s illness [1]. This way of practicing medicine shows, not only patients’ satisfaction, but also the positive impact on patient’s health treatment adherence and optimization of health care costs [2,3,4,5]. Bombeke et al. [6] found that students, despite expressing positive attitudes toward patient-centeredness teaching models, feel that they do not receive sufficient training during their medical studies to attain the level of competent behavior needed in today’s challenging hospital environment. When it comes to dealing with time pressure, tiredness, and non-patient-centered role models, they lose focus. Finally, they concluded that a supportive student–doctor relationship and student-centered education and guidance that addresses the needs of the doctor-as-person are central to the development of patient-centeredness.

Education experts state that palliative care education (PCE) enables undergraduate students to act professionally, helping them to acquire skills that are crucial for their professional development [7,8]. Challenges to student professionalism may arise from emotionally charged situations, such as handling patients and families during death and dying experiences. Appropriate behavior mastered in these circumstances may likewise be applied to other emotionally challenging situations [9]. When the students reflect about palliative care practice, they do not only learn about symptom control and end-of-life care, but they interact with interdisciplinary teams, where they are able to find role models of how to deal with complex psychological, social, ethical, cultural, and religious situations [10,11,12,13,14,15,16,17].

Our team has achieved a deep understanding about how PCE could be a good patient-centered teaching model and we have published four qualitative studies that focus on students’ experiences related to palliative care (PC) teaching interventions. The first one was a qualitative exploratory study on narrative reflections of nurse students after a PC optional course; most of the students concluded that the course was an essential component in their training, which contributed to a personal and professional development [18]. The second was done following the same methodology, but the narratives about the PC course were done by medical students, and they expressed an overall improvement of their vision of medicine [19]. The third study analyzed the narratives of medical students, but this time about a bedside PC clinical experience, this experience promoted a deep reflection, that provided for most of the students a deeper understanding of core values of medical practice, and some of them even described this bedside experience as life-changing [20]. Finally, we published the analysis of a reflective exercise of a PC end-of-life decision-making course for medical students, performed at a modern art museum, the results of which showed that PCE helps medical students address several competencies related to being patient-centered and empathic [21]. The thematic analysis of these previously published studies done by our group shows a wider range of aspects than those measured in the actual questionnaires that assess PCE, focused on measuring fears and attitudes towards end-of-life patients care, or symptom control knowledge [22,23,24,25,26]. On the basis of the overall thematic analysis of these studies, we realized that it was possible to obtain an inventory that covers the five assessment areas defined by Wilkinson as a blueprint for assessing professionalism [27]. This researcher provided, after carrying out a systematic review on this topic, a clustered definition of professionalism into the following five assessable components: adherence to ethical principles, effective interactions with patients and people who are important to those patients, effective interactions with other agents working in the system, reliability and commitment to self-maintenance, and gaining of competences. We also explored the existence of a published questionnaire that had already assessed the professionalism cluster defined by Wilkinson. However, the ones we found, when compared with Wilkinson´s framework, only addressed partial areas, were too general, or lacked well-explained validation processes [28,29,30].

To meet the need of a questionnaire that allows undergraduate students to self-report their ability to perform patient-centeredness medicine, and to assess the improvement of their attitudes towards end-of-life teaching interventions, we developed an original inventory based on nurse and medical students’ reflections after PCE interventions. The aim of this study is to design and validate a student’s self-report professionalism inventory based on PCE.

## 2. Methods

The design and validation process of the inventory followed a mixed method with sequential exploratory strategy. This strategy is useful when developing and testing a new instrument [31]. The Research Ethics Committee at the University of Navarre approved this study (reference: P125/2016). Participants signed informed consent forms. Data were collected and managed anonymously. 

### 2.1. Design of the Inventory 

The inventory was built on the 150 themes obtained in the thematic analysis of four qualitative studies [23,24,25,26] on undergraduate palliative care education interventions carried out and published by our group. With those 150 themes we wrote 150 possible items for the inventory. Three researchers (AN, CC, MA) analyzed each item through constant comparison: duplications were eliminated, and items with similar statements were gathered, obtaining the basis for the first version of a 55-items inventory with 0–10 scales to evaluate agreement or disagreement with the item statement.

### 2.2. Population

Almost three hundred medical students from the University of Navarra participated in the two validation studies. In the first survey, 4th year students completed the survey concerning two diverse educational activities (66 before the thirty days educational activity on compassion and 62 before receive usual one day lectures on clinical topics; and 34 and 30 respectively, after the activities); median age 21, 65% female; 30% were from Navarra and 70% from other regions of the country or foreigners (4 were international students from Spanish-speaking countries).

In the second survey, 6th year students filled in the inventory at the beginning of their last year of medical education, and when they finished their palliative care curriculum (median age 23, 63% female and from all over Spain), 197 students of 206 implemented the inventory at the beginning of the academic course. 

### 2.3. First Survey with 128 Medical Students 

In a first survey, as we mentioned, the questionnaire was applied to a sample of 128 medical students in the 4th academical year: half participated in an educational intervention on professionalism and half before they received usual one day lectures on clinical topics.

The teaching intervention on professionalism started with a 4 h compassion workshop and was follow by a clerkship and a clinical case on compassion observed. In the first half of the workshop there was a brainstorming on the concept followed by the discussion of two clinical scenes from the mainstream film “Whit”. In the second half, lived experiences of a professional, a patient, and a relative were presented followed by free questions from the students. This workshop was just a few days before the first clinical clerkship at the hospital and is part of the curriculum of the University of Navarra within a program to enhance professional identity. After the 4 weeks clerkship, students present an evaluation of a written clinical case on compassion. In this way, the teaching intervention pretended to start and induce the 1 month reflection on this concrete professional value.

With the response of the students, psychometric properties of the inventory were studied as follows: 

(1) Reliability: internal consistency with Cronbach’s alpha, t-test stability with intraclass correlation coefficient in a 24 h interval

(2) Construct validity: hypothesis testing with the two different educational interventions (Mann–Whitney U test). As no similar questionnaire could be found to compare changes after the intervention, a hypothesis was applied, whereby an increased score on the inventory was to be expected after a teaching activity, being both higher and statistically significant to the change observed after a day of usual clinical lectures [32].

(3) Responsiveness to the professionalism teaching intervention was tested with the Wilcoxon test. 

(4) Structure: (a) An item selection was done through the initial factor analysis performed with Varimax rotation; (b) exploratory factor analysis (EFA) using Promax power rotation was used to identify sub-scales, and the Kaiser–Meyer–Olkin test (KMO) was done to assure that the inventory was suitable for EFA.

### 2.4. Second Survey with 164 Medical Students 

In a second survey, the 33-items questionnaire was applied to a sample of 164 students in their 6th year at the beginning of the academic course. Structural validity (the confirmatory factor analysis structural equations model and goodness of fit statistics, expecting a comparative fit index (CFI) between 0.8–1, and a standardized root mean square index (SRMR) close to 0.08) [33], and internal consistence (Cronbach´s α coefficient) of the 33-item inventory, was performed with the response of the students. The feasibility of the inventory was also tested in this group measuring the time for completion and the response ratio. 

The final inventory structure was qualitatively compared by two researchers (AN, MA) with the Wilkinson assessment areas framework for professionalism [27]. Wilkinson proposal consisted in five themes: (1) Adherence to ethical practice principles, (2) effective interactions with patients and with people who are important to those patients, (3) effective interaction with other people working within the health system, (4) reliability, and (5) commitment to autonomous maintenance and continuous improvement of competence.

## 3. Results

The inventory is a list of 33 items presented in only one sheet (Table 1 shows an English version of the SIP to facilitate understanding, as translated by an English native speaker with the review of the authors. The original and validated version is offered in (Appendix A) in Spanish). That list of items represents the end point of a process of synthesis on more than 600 students’ reflections about the impact that palliative care medical teaching has had on their professionalism. The inventory was tested for psychometric properties with two different samples of 128 and 164 medical students in the two successive versions of 55 and 33 items, respectively. Each item is presented written as a statement in first person to be read, followed by a scale 0–10 expressing the auto-evaluation of the respondent on a level of agreement with the statement (0 = no agreement at all, 10 = total agreement). The items are grouped into seven sub-scales following the results of the factorial analysis: holistic approach (11 items), caring for and understanding the patient (6 items), personal growth (4 items), teamwork (3 items), decision-making (6 items), patient assessment (6 items), and being a health care professional (1 item). We call the new assessment tool “Student’s Inventory on Professionalism” to indicate with the name the construct explored and that it is grounded in students’ perceptions. Its acronym “SIP” suggests how fast and easy it can be completed.

Table 2 shows an overview of all the psychometric characteristics of the inventory. Construct validity was confirmed with the hypothesis test. The difference in average scores in the group, which took part in the compassion teaching activity, was 17 points ±22, compared to the difference of 5 points ±13, given the values, which appear in the group that attended usual clinical lectures (test re-test Mann–Whitney, *p* < 0.02). Intra-observer stability showed a good intraclass correlation with all the values over 0,4. The inventory also showed responsiveness to the professionalism teaching activity. Out of a total of 320 points, the average points total went from 238 ± 31 to 255 ± 30 after the teaching activity, with a significant increase of 17 points (Wilcoxon test, *p* < 0.001). After the intervention, an increase in the score was found in 28 of the 33 questions, a statistically significant increase and at least one whole point increase out of ten was found in 9 of the 33 questions. The inventory was easy to complete for all the students without comprehension difficulties and in less than fifteen minutes.

In the design process and with the aim to reduce the length of the tool, an initial factorial analysis was carried out with the 4th year student’s response to the 55-item proposal showing 14 factors. The method used was principal-components analysis with orthogonal varimax rotation. Factor 1 explained 70% of the whole variability of the inventory, with a saturation load above 0.35 for all the scales. This means that the inventory succeeded in measuring a common conception. After the factor analysis of the 55-item proposal, it was not possible to group questions following proper categories because we got 14 factors and all of them with valid eigenvalues over 1. Then, a careful analysis allowed us to discard four items due to saturation below 0.4 for Factor 1. Qualitative semantic analysis of the 51 remaining items, pairing those with similar wording and/or meaning, accomplished by three researchers, allowed to reduce the entire set to 32 statements. As an item asked two concepts at the same time, if the student was able to seek for “help” and “advice”, we decided to unfold this item in two questions. Minor changes were introduced in some of the items for better comprehension and, finally, we got the 33-item inventory. 

SIP categories were obtained with an exploratory factor analysis of the final 33-item inventory version (ProMax power rotation) showing nine factors that were grouped on seven categories, each category expressing a construct with a name (three factors were joined in one subscale because of content similarities) (KMO = 0.81). All the items had a reasonable inter-item reliability; six factors obtained a Cronbach’s alpha between 0.64 and 0.88. As a factor is only one item, it is not applicable for Cronbach’s alpha analysis. The confirmatory factor analysis comparative fit index (CFI) of the seven latent constructs using a structural equation model was 1 with a Standardized Root Mean Square index 0.088 (SRMR), all the items explained between 0.45 to 0.89 of the value of the construct where it is located, and the equation goodness of fit statistics showed a 0.99 overall R-square coefficient for the inventory with appropriate coefficients for each item (Table 3). The correlation between different constructs of the inventory showed how the final seven inventory dimensions are related and at the same time different enough to measure the cited variables (Table 4). The final seven inventory dimensions presented a good reliability, final Cronbach’s alpha values showed a good internal consistency, even the values of the constructs obtained with the 6th year students of the second sample (response variation 0.73 to 0.84) were better than the ones obtained with the seven dimensions analyzed with the first sample response.

All components of professionalism described by Wilkinson et al. are covered in the inventory (Table 5). SIP addresses, with several items, the component of adherence to ethical principles and the effective interaction of the professional with the patient and colleagues. The exploration of the commitment to improvement is also ensured as well as the reliability. The only elements that the students left out in their reflections and are related to professionalism are those that correspond to practical or very specific aspects of professional behavior, such as to how to assume responsibility, stay within professional limits, exercise leadership, or exercise in the audit.

## 4. Discussion

The result of this study was the design and validation of the first version of an original inventory for students´ self-report of professionalism. It has been designed following health students´ learning experiences on patient-centeredness, end-of-life, and palliative care education interventions [18,19,20,21] and tested on a professionalism education intervention focused on medical compassion.

Content validity emerges with the process of developing the inventory. Data qualitative selection followed by data quantitative selection resulted in the actual SIP version. The studies on which this inventory is based, and other similar works [34,35,36], provided students with a self-perception of professionalism learning. Thereby, they create nothing more than hypotheses, which cannot be presented as tested or generalizable. After the solid process of creating the inventory presented in this study, these hypotheses are ready to be subject to empirical tests.

Professionalism is a wide concept, with contextual and cultural differences. Except Penn State College of Medicine Professionalism Questionnaire (PSCOM) [28], no other tool assesses all the areas that this concept covers [28,29,30,37]. Furthermore, the items contained in the SIP consider aspects such as caring and compassion, sensitivity, tolerance, openness, communication, respect for patient dignity and autonomy, respect for other health care professionals and staff including teamwork, relationship building, motivation, insight, commitment, excellence and scholarship, and so forth, that cover the five areas described by Wilkinson et al. as a blueprint to assess professionalism [27]. Therefore, the SIP would provide a valid method for evaluating whether medical education interventions centered in the student as a person contribute to the development of professional identities among future health professionals. The SIP also could test the indirect impact on the students of non-specific professionalism teaching interventions.

The SIP, as a self-administered scale, assesses attitudes and can aid reflection, but cannot assess what a student actually does. When compared with other self-administered scales, the SIP has a more patient-centered approach than the PSCOM, which focuses more on organizational areas [28]. To assess what students actually do, it is better to use tools applicable during clinical encounters, such as the mini clinical evaluation exercise (mini-CEX) [38], or the ones that ask for the patient’s opinion [39], but, in the end, all of them are limited to what happens during the clinical encounter and cannot evaluate professionalism areas related to commitment to improvement.

In the psychometric analysis carried out, the SIP was a valid tool shown to be reliable and sensitive for measuring the efficacy of professionalism in educational activities. The SIP structure’s preliminary data provides a preliminary support to the inventory and factorial analysis with a confirmatory factor analysis on a new sample contributed to reinforcing it. As the 33 items are clustered into seven sub-scales conformed by different conceptual areas, in the future, it will be possible to test whether those areas behave like genuine sub-scales, maintaining internal consistency and providing partial scores which make it possible to compare different professionalism training activities and different lecturers or universities. As a limitation of the statistical analysis of the structure of the inventory, we have to admit that the sample of the first survey was only 128 inventories, not sufficient to obtain a reliable and accurate result for an exploratory factor analysis, that was the main reason why we tested the inventory again on a second sample of 164 inventories, although the first survey obtained a KMO of 0.81.

To analyze SIP construct validity, we employed the hypothesis test, assuming that an educational intervention on compassion would increase the score on the scale. We have found that the initial average of all questions was high (nearly all the scores were above six). The behavior of the inventory could be biased by the vocational component among medical students at the end of the compassion teaching intervention, but it is also possible that the high scores prior to the intervention were due to a complacency bias or ceiling effect. It is also difficult to interpret properly the inventory responsiveness, because there was not an initially established value to indicate what would be considered as a significant increase in the attitudes explored. Moreover, the inventory was first tested on a single university, with a strong teaching in humanities; this also could contribute to the initial responses´ high average. Those effects could be reduced in future interventions, if the SIP is employed at the end of the teaching session, asking to score each item on self-perception, before and after the intervention.

The SIP responsiveness was as expected. The scores obtained after 24 h with the re-test were totally stable. In addition, the results, after the professionalism teaching intervention, were uniformly positive. Furthermore, the internal consistency obtained by the SIP confirms this reliability, this consistence was improved through the psychometric and semantic data selection. The factorial analysis applied, ensures, at the same time, that different items contribute with unique, identifiable information. 

The results of the validation strategy used, including a teaching experience, allow us to assert that the SIP promises to be a sensitive tool for detecting the effects of teaching activities in professionalism. If the inventory detected statistically significant changes in one out of every three items, with a very small sample and after a very short five-hour intervention, it would be expected that this same behavior would be maintained with more consistent interventions for more numerous groups where the statistical power is not as compromised. As another limitation**,** the research team cannot deny that the use of the first version of this inventory would lead to modifications to complete its capability as an on-going process of continuous validation [40].

## 5. Conclusions

The SIP is a new and promising inventory, with a reasonable internal consistency and reliability, and sensitive to assess the effectiveness of global educational interventions in professionalism. These properties lead us to think that it is a suitable inventory for future prospective studies related to palliative care and professionalism education interventions. These studies also would help with the necessary further validation of the inventory.

The SIP will allow to test if the initial student’s reaction of enthusiasm and surprise with their PCE is constant over time, after going through years of medical practice. It would also help to perform comparative analyses on the performance of educational activities in different professions and between different universities. Including sociodemographic variables in future studies would allow us to study which other personal and cultural factors influence professionalism learning.

## Figures and Tables

**Table 1 ijerph-16-04925-t001:** Items and subscales of the Students’ Inventory of Professionalism (SIP).

Sub-Scale	#	Item
Holistic Care	1	I have learned to listen to patients
	2	I have learned to show humanity to the patient
	3	I have learned to devote time to each patient´s needs
	4	I have learned to give emotional and spiritual support to my patients
	5	I feel capable of adapting to each patient
	6	I feel capable of giving hope to the patient, without creating false expectations, when discussing the progress of their illness
	7	I feel capable of helping the patient maintain their dignity despite their deteriorating condition
	8	I feel capable of helping my patients’ families and giving them support for their needs
	9	I have learned how to gain patients’ trust
	10	I have learned how to give bad news to patients in a caring manner
	11	I feel capable of managing my feelings appropriately when treating patients in complex situations
Care and understanding	12	I have learned that caring is the essence of my profession
	13	I have learned how to be close to my patients
	14	I have learned that when a patient has a disease with a poor/incurable condition, still, it is always possible to do something
	15	I have learned to take into account the patient’s previous experiences in order to better understand him or her
	16	My clinical experience has helped me to understand how the patients adapt to their illness
	17	I feel capable of listening to others before making a difficult decision
Personal Growth	18	My clinical experience has helped me to grow as a person
	19	My clinical experience has helped me to better understand my professional colleagues
	20	My clinical experience has helped me to maintain (or recover) my initial commitments to developing as a healthcare professional
	21	I have learned to value the gratitude I receive from patients with an advanced disease
Teamwork	22	I feel capable of working in a team
	23	I have learned how to seek advice from colleagues when needed
	24	I have learned to seek help when I need it
Decision-making	25	I feel capable of involving the patient and their families in making decisions at the end of life
	26	I reassess my treatment plan when it causes suffering
	27	I have learned to keep an open-mind and not to judge the patient on first impressions
	28	I have learned to respect the patient’s wishes
	29	I have learned to individualize the decision-making process
	30	I have learned that each treatment must be personalized
Patient assessment	31	I feel capable of undertaking appropriate symptom evaluation
	32	I feel capable of performing a multidimensional, holistic assessment of patients
Being a professional	33	I have learned that my profession is science with compassion

**Table 2 ijerph-16-04925-t002:** Psychometric properties of the Students’ Inventory of Professionalism (SIP).

Property	Analysis	n	Sample	Statistical Test	Result
Reliability	Initial Internal consistency (first sample)	128	First Survey	Cronbach’s Alpha for factor 1 to 5 (Scale reliability coefficient for factor 6 Factor 7 only one scale)	Factor 1 = 0.88 Factor 2 = 0.83 Factor 3 = 0.76 Factor 4 = 0.64 Factor 5 = 0.65 Factor 6 = 0.33 Factor 7: Not applicable
	Stability: Test-retest	28	First Survey	Intraclass coefficient of correlation (ICC)	Global ICC r = 0.95; All Items ICC r > 0.4.
	Final Internal consistency (second sample)	164	Second Survey	Cronbach’s Alpha for construct 1 to 5 (Scale reliability coefficient for construct 6 Construct 7 only one scale)	Construct 1 = 0.84 Construct 2 = 0.78 Construct 3 = 0.73 Construct 4 = 0.83 Construct 5 = 0.81 Construct 6 = 0.84 Construct 7: Not applicable
Construct Validity	Hypotheses-testing	32 vs 28 (1)	First Survey Two subgroups: pre-post different educational activity	Mann-Whitney U, to test differences in mean values	Mean difference: - Intervention group 17 ± 22 - Control group 5 ± 13, *p* < 0.002
Responsiveness	Intervention testing	32 (1)	First Survey: pre-post professionalism teaching intervention	The Wilcoxon signed-rank test	Mean difference = 17 238 ± 31 pre-intervention to 255 ± 30 post-intervention *p* < 0.001
Structure	Items selection to reduce sample	128	First Survey	Factor Analysis with Varimax rotation	14 factors. Factor 1 explains 70% of the whole variability with loads greater than 0.35 for each variable. Reduce 55 to 33 Items
	Exploratory Factor Analysis to determinate sub-scales	128	First Survey	Factor Analysis with Promax power rotation	Nine factors were chosen explaining 68% of the total variability of the 33 variables. Kaiser-Meyer-Olkin (KMO) = 0.81 (adequate sample) and were resumed in seven dimensions
	Confirmatory Factor Analysis	164	Second Survey	Structural Equation Model	Equation Goodness of Fit Statistics 0.99

(**1**) Only 28 pairs were valid for the inter-observer reliability analysis and 32 for the responsiveness analysis, due to difficulties in the pairs identification process.

**Table 3 ijerph-16-04925-t003:** SIP confirmatory factor analysis of the seven latent constructs (*n* = 164, medical students, 6th year).

Subscale	Item	Average	SD	SC (1)	R-Squared (2)	Inter-Item Reliability		
					Overall = 0.99	If item is removed	Item-rest correlation	Cronbach’s alpha
Holistic Care	1	8.5	1.4	0.50	0.26	0.84	0.37	0.84
	2	7.7	1.8	0.46	0.22	0.84	0.28	
	3	7.1	2.1	0.55	0.31	0.83	0.45	
	4	6.2	2.4	0.68	0.46	0.81	0.61	
	5	7.1	1.8	0.67	0.45	0.82	0.55	
	6	6.2	2.1	0.64	0.42	0.81	0.71	
	7	7.7	2.2	0.65	0.42	0.82	0.53	
	8	6.9	2.1	0.71	0.50	0.81	0.63	
	9	6.5	2.1	0.70	0.49	0.81	0.63	
	10	4.6	2.6	0.44	0.20	0.82	0.51	
	11	5.6	2.5	0.53	0.28	0.83	0.46	
Care and understanding	12	8.3	1.9	0.61	0.38	0.75	0.50	0.78
	13	8	1.7	0.70	0.49	0.74	0.56	
	14	7.7	2.1	0.56	0.31	0.75	0.51	
	15	7.4	1.7	0.67	0.45	0.73	0.61	
	16	6.7	2.1	0.70	0.50	0.75	0.50	
	17	8.4	1.6	0.58	0.34	0.75	0.50	
Personal Growth	18	8.6	1.4	0.75	0.58	0.67	0.54	0.73
	19	7.9	1.7	0.72	0.53	0.64	0.58	
	20	8.2	2.1	0.67	0.46	0.67	0.54	
	21	7.7	2.2	0.59	0.36	0.70	0.50	
Teamwork	22	8.7	1.3	0.71	0.51	0.87	0.61	0.83
	23	8.6	1.4	0.89	0.80	0.68	0.80	
	24	8.6	1.4	0.83	0.69	0.74	0.72	
Decision-making	25	7	2.6	0.56	0.32	0.80	0.55	0.81
	26	7.2	2.4	0.57	0.33	0.78	0.60	
	27	7.6	1.9	0.60	0.37	0.80	0.47	
	28	7.9	1.8	0.78	0.61	0.76	0.67	
	29	7.7	2	0.83	0.70	0.75	0.71	
	30	8.6	1.4	0.65	0.43	0.80	0.51	
Patient assessment	31	6.9	1.8	0.79	0.63	Average interitem covariance 2.52		0.84
	32	6.8	1.9	0.95	0.92			
Being a healthcare professional	33	8.4	1.6	0.89	0.81	NA		

(1) Confirmatory factor Load or Standardized Coefficients (SC) of the structural equations model: express to what extent each item explains the construct or subscale within which it is included. (2) Squared multiple correlation coefficients (R-squared): express goodness of fit statistics for non-recursive systems that involve endogenous variables with reciprocal causations.

**Table 4 ijerph-16-04925-t004:** Correlation between constructs of the inventory.

Correlation between Sub-Scales (1)	HC	CU	PG	TW	DM	PA
Holistic Care (HC)	-					
Care and Understanding (CU)	0.85	-				
Personal Growth (PG)	0.58	0.76	-			
Team-work (TW)	0.36	0.41	0.48	-		
Decision-making (DM)	0.63	0.74	0.54	0.46	-	
Patient assessment (PA)	0.47	0.48	0.53	0.28	0.57	-
Being a healthcare professional (BHP)	0.28	0.53	0.57	0.23	0.52	0.52

(1) The correlation was statistically significant (*p* < 0.001) among all the subscales using the Structural Equations Model.

**Table 5 ijerph-16-04925-t005:** How elements of professionalism (Wilkinson et al., 2009) matched with items of the Student’s Inventory of Professionalism.

Assessable Components of Professionalism	Elements	Item Related in SIP Inventory
Adherence to ethical principles	Honesty and integrity	2, 12, 33
	Confidentiality	9
	Moral reasoning	7
	Respect	27, 28
Effective interaction with patients	Respect for diversity	4, 15, 5
	Politeness	6
	Empathy	1, 10, 13
	Manner	31, 32
	Include patient in decision making	25, 8
	Maintain professional boundaries	-
	Teamwork	22
Effective interaction with coworkers	Balance availability with oneself care	23

	Manner/demeanor	24
	Respect	17
	Politeness/patience	-
	Maintain professional boundaries	-
Reliability	Organized	2
	Accountability	29, 30
	Take responsibility	-
Commitment to improvement	Personal awareness, reflectiveness	11, 16
	Lifelong learning	18, 19, 21
	Advance knowledge	20
	Deal with uncertainty	14, 26
	Provide feedback	-
	People management/Leadership	-
	Seek and respond to an audit	-

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
