# Peer review of "Student’s Inventory of Professionalism (SIP): A Tool to Assess Attitudes towards Professional Development Based on Palliative Care Undergraduate Education"

_ijerph, 2019, doi:10.3390/ijerph16244925_

Round 1
Reviewer 1 Report
I commend the authors on designing a much needed scale for this field. With that said I believe that this is a good paper and can be strengthened with some editing to help improve the quality of the paper. My main concerns are with the organization of the background and the methods section which I believe is not clear enough to describe the project sufficiently for readers particularly since this paper is all about designing a scale/instrument. This paper has a lot of potential and I encourage further work.
Abstract as well as for the entire paper please check for grammatical errors. If possible make the Abstract more succinct and straight to the point e.g. re-structure the line 31 starting with reliability (you do not need to list all forms of reliability. Introduction needs re-writing - strengthen with more background. Cite properly as per journal standards. Here are a few examples. Line 51 is unclear. Please rewrite Line 54 – 57 -please rewriteLine 58: Wilkinson in 2009 provided a blueprint for assessing professionalism. Cite properly according to journal standards For the last paragraph in the introduction, I am not sure if the authors are stating that they performed four qualitative studies prior to this study or if the four studies are part of this particular study for the paper. Please be explicit so as not to confuse the reader. 3. I do not see a section with the characteristics of the population. Four studies were done so we should see demographics participants for each study, if the studies were done specifically for this paper. Regardless, we should see the demographics of the sample used to test the survey. 4. If the inventory was built on 4 published studies perhaps the introduction should include more about these 4 published studies. Please put a special section in the methods section to talk about the 4 studies. 5. A more thorough explanation of the intervention is needed 6. Please mention how participants were recruited. How many participants were recruited? how many answered survey? How were the participants recruited? 7. More information about the qualitative analysis:Content or
thematic analysis should describe the coding process in detail, including who was involved, open- and focused-coding, and in the case of multiple raters, how disagreements were resolved and measures of inter-rater reliability. There are a number of other practices that the authors should report to ensure reliability and validity in findings from qualitative research, including: triangulation of data sources, member checks, saturation, reflexivity, peer review, and audit trails. Please provide information on as many of these as possible in the analysis section for the qualitative portion of the project. While the authors did attempt to mention all of this, they were not explicit enough. Please review paper to make it more psychometrically sound.
Author Response
First of all, we really appreciate the comments and insights about the manuscript, I am sending on behalf of all the authors a new version because the changes proposed to do.
Abstract as well as for the entire paper please check for grammatical errors. If possible make the Abstract more succinct and straight to the point e.g. re-structure the line 31 starting with reliability (you do not need to list all forms of reliability.
Abstract: as proposed, we have prepare a more succinct version.
Introduction needs re-writing - strengthen with more background. Cite properly as per journal standards. Here are a few examples. Line 51 is unclear. Please rewrite Line 54 – 57 -please rewriteLine 58: Wilkinson in 2009 provided a blueprint for assessing professionalism. Cite properly according to journal standards For the last paragraph in the introduction, I am not sure if the authors are stating that they performed four qualitative studies prior to this study or if the four studies are part of this particular study for the paper. Please be explicit so as not to confuse the reader. 3. I do not see a section with the characteristics of the population. Four studies were done so we should see demographics participants for each study, if the studies were done specifically for this paper. Regardless, we should see the demographics of the sample used to test the survey. 4. If the inventory was built on 4 published studies perhaps the introduction should include more about these 4 published studies.
Introduction: we have included a brief explanation of the four qualitative studies, already published by our research team, that are the base of the inventory. Moreover, we have reorder the introduction in order to make it more easy to understand.
Please put a special section in the methods section to talk about the 4 studies. 5. A more thorough explanation of the intervention is needed 6. Please mention how participants were recruited. How many participants were recruited? how many answered survey? How were the participants recruited? 7. More information about the qualitative analysis:Content or thematic analysis should describe the coding process in detail, including who was involved, open- and focused-coding, and in the case of multiple raters, how disagreements were resolved and measures of inter-rater reliability. There are a number of other practices that the authors should report to ensure reliability and validity in findings from qualitative research, including: triangulation of data sources, member checks, saturation, reflexivity, peer review, and audit trails.
Method: we think it is not necessary to add an explanation of the four qualitative studies about Palliative Care teaching interventions, because we have clarify that have been already published. We have clarify the process of validation both in methods and results.
Please provide information on as many of these as possible in the analysis section for the qualitative portion of the project. While the authors did attempt to mention all of this, they were not explicit enough. Please review paper to make it more psychometrically sound.
In order to explain better population, we have add a paragraph in results.
Thanks for your precious time reviewing this manuscript
Reviewer 2 Report
The paper on developing a tool on professionalism is very helpful, and there are few other tools.
The statistical methodology described to develop the tool is quite complex and it is not my area of expertise to comment on this. However, it did seem a sound iterative process.
Overall, the English grammar is very good, but there are many examples where there are small errors in the grammar.
Line 26 our society requires.
Line 62 competencies
Line 83 perform patient-centred medicine,
Line 96 there was a positive impact on professionalism
103 128 medical students in the 4th academic year
112 evaluation of a written clinical case on compassion
129 academic course
141 The inventory had a list of ...
Table 1 presents an English version of Inventory
147 The inventory was tested for psychometric properties with two different
157 I have learned to devote time to each patient's needs
162 before they recieve usual one day lectures on clinical topics
Grammar also needs correction in lines 177, 192, 225 and 275.
Author Response
We are grateful to your comments that will increase for sure the quality of the manuscript.
We have made some changes in order to explain better the validation statistical process.
The paper on developing a tool on professionalism is very helpful, and there are few other tools.
The statistical methodology described to develop the tool is quite complex and it is not my area of expertise to comment on this. However, it did seem a sound iterative process.
Overall, the English grammar is very good, but there are many examples where there are small errors in the grammar.
Line 26 our society requires.
Line 62 competencies
Line 83 perform patient-centred medicine,
Line 96 there was a positive impact on professionalism
103 128 medical students in the 4th academic year
112 evaluation of a written clinical case on compassion
129 academic course
141 The inventory had a list of ...
Table 1 presents an English version of Inventory
147 The inventory was tested for psychometric properties with two different
157 I have learned to devote time to each patient's needs
162 before they recieve usual one day lectures on clinical topics
Grammar also needs correction in lines 177, 192, 225 and 275.
We have included in the manuscript all the gramatical changes proposed.
Thanks for the time expend in this revision.
Round 2
Reviewer 1 Report
Thank you so much for doing the requested changes. The paper is much more improved, just a few minor grammatical errors which needs reviewing.
The authors did not feel the need to explain the process of the four previous studies done, but did a good job in incorporating it into the introductions (line 66) - Thank you for doing that.
In the abstract Line 30 instead of saying:
Those studies were already published by our research team. The authors can say (see author, date/reference number) e.g. see Noguera1, Arantzamendi2, Lopez-Fidalgo3 and Gea4).
For example: The inventory was built based on the themes that emerged from the analysis of four qualitative studies about nurse and medical students’ perceptions related to palliative care teaching interventions (see Noguera1, Arantzamendi2, Lopez-Fidalgo3 and Gea4).
Line 45; Health faculties should provide to their future professionals
Line 66 - spelling error
Line 83 - grammatical error
Line 85 - citation error (missing reference number)
Line 91 and 92 (citation error - missing reference number.
Note: Also line 160 - 70% from ... (e.g other locations, states, countries) replace the word elsewhere
Line 159 - students were, as median age 21 years old and 65% female; 30% were from Navarra and70% from elsewhere (including 4 were international students).
(median of age 23 years old, 63% female and from all over different (e.g. states, counties etc in Spain),
Review the rest of the article for minor errors like this
Author Response
Thanks again for the effort and time put in the revision.
We have made the changes suggested in the abstract and in the different parts of the manuscript.
We have revised all the references, a problem probably related with different software versions, corrections have been made.
We also have revised the text to correct gramatical errors.
Kind Regards